# Innovative Approaches for Food: Using Natural Phenolic-Rich Extracts to Produce Value-Added Fresh Pasta

**DOI:** 10.3390/ijms241512451

**Published:** 2023-08-05

**Authors:** Sandra M. Gomes, Daniela Albuquerque, Lúcia Santos

**Affiliations:** 1LEPABE—Laboratory for Process Engineering, Environment, Biotechnology and Energy, Faculty of Engineering, University of Porto, Rua Dr. Roberto Frias, 4200-465 Porto, Portugal; scgomes@fe.up.pt; 2ALiCE—Associate Laboratory in Chemical Engineering, Faculty of Engineering, University of Porto, Rua Dr. Roberto Frias, 4200-465 Porto, Portugal; 3Faculty of Engineering, University of Porto (FEUP), Rua Dr. Roberto Frias, 4200-465 Porto, Portugal; up201904810@edu.fe.up.pt

**Keywords:** natural products, bioactive compounds, antioxidants, food fortification, cereal-based products, fresh pasta

## Abstract

Cereal-based products, which are rich in carbohydrates, are widely consumed worldwide; however, this type of food lacks other nutrients. Phenolic compounds from natural sources, such as *Moringa oleifera*, can be incorporated into these products to increase their nutritional and biological value. In this study, a phenolic-rich extract was obtained from *M. oleifera* leaf powder using a Soxhlet extractor. The extract obtained presented a total phenolic content of 79.0 mg of gallic acid equivalents/g and the ABTS and DPPH assays showed that the extract can act as an anti-oxidant agent, with IC_50_ values of 205.2 mg/L and 636.0 mg/L, respectively. Afterwards, fresh pasta was produced and the extract was incorporated into the pasta to improve its biological properties and extend its shelf-life. The results demonstrated that the addition of *M. oleifera* to the fresh pasta increased its anti-oxidant capacity and did not interfere with the cooking properties of the product. Moreover, the fortified pasta presented an increased shelf-life, since the extract conferred protection against microbial contamination for longer periods of time. Therefore, these findings showed that the incorporation of phenolic-rich extracts from natural sources (such as *M. oleifera*) is a feasible sustainable biotechnological approach to produce value-added cereal-based products.

## 1. Introduction

Cereal-based products are widely consumed around the world, playing an important role in the diet of many cultures and providing a good source of carbohydrates [1]. Pasta, for example, is a staple food loved by and familiar to all for its versatility and affordability. However, despite its high content of carbohydrates, pasta lacks essential amino acids and micronutrients, such as minerals and vitamins, as well as bioactive compounds (BACs). This makes pasta an interesting vehicle for functional ingredients [2,3]. Given the growing interest of today’s society in a healthy lifestyle, the fortification of foods with natural ingredients of great nutritional importance has become more popular and, consequently, its research has been stimulated. Many functional ingredients are derived from natural sources that naturally contain compounds with beneficial properties for human health [4,5,6,7].

*Moringa oleifera* is a widely cultivated plant from the Moringaceae family that presents great potential to be used as a natural additive to enhance food’s nutritional profile and health benefits. This tree is native to tropical and subtropical areas, such as the sub-Himalayan tracks of India, Pakistan, Bangladesh, and Afghanistan [8]. It is universally referred to as “miracle tree” due to its rich composition in nutrients and BACs of great importance to human nutrition and health. Most parts of the plant, including roots, seeds, flowers, and leaves, are edible and contain numerous nutrients such as proteins, carbohydrates, vitamins (A, B1, B2, B3, C, and E), fibre and minerals. Moreover, the plant is reported to contain a great number of BACs, namely phenolic compounds, tannins, and carotenoids [8,9,10,11]. The appealing composition of *M. oleifera*, allied with the fact that it grows quickly and requires little agricultural care, as it can strive in drought and underprivileged soil, makes this tree an interesting sustainable option for food fortification [12]. 

To ensure the consumers’ safety, it is important to guarantee that no adverse effects occur upon consumption of the fortified products. Several studies were performed to access the toxicity of *M. oleifera* extracts [13]. Table 1 summarises the results obtained from some of these reports. 

Overall, based on the findings of different studies, *M. oleifera* extracts seem to be safe for human consumption in the amounts commonly used and, to date, no negative impact has been reported in human studies [13,18].

Nowadays, consumers are attracted by food products presenting bioactive properties. Due to the presence of a broad spectrum of BACs, *M. oleifera* has powerful anti-oxidant, anti-inflammatory, hypoglycaemic, hepatoprotective, antimicrobial, antimutagenic, and anticancer properties [8,9,10,19]. *M. oleifera* leaves are especially rich in phenolic compounds, such as phenolic acids (e.g., gallic acid, chlorogenic acid, caffeic acid) and flavonoids (e.g., rutin, quercetin, kaempferol), which are secondary metabolites with one or more hydroxy groups responsible for its high antioxidant activity [10,20,21,22,23,24]. Therefore, the use of *M. oleifera* as a food ingredient is receiving a great deal of attention, not only because of its nutritional value but also due to its high anti-oxidant capacity, which, in addition to bringing benefits to human health, delays the oxidative processes associated with food degradation, increasing its shelf-life [25]. Many studies have shown its potential use in cereal-based food products as a natural additive to improve their nutritional value and proprieties (Table 2).

Several studies have proven the suitability of *M. oleifera* in the fortification of various cereal-based food products. However, to the best of the authors’ knowledge, no studies were performed using *M. oleifera* extracts, which present a higher concentration of BACs when compared to the powder. Also, although good results were obtained regarding the improvement of the nutritional values and properties of the products, few studies evaluated the effect on their antioxidant properties, which are important for both human health and the product’s shelf-life. This last aspect is particularly important in the case of fresh pasta, which presents a short shelf-life of 2–3 days [34]. 

Therefore, the aim of this study was to produce fresh pasta fortified with *M. oleifera* leaf extract and assess its effect on the physicochemical properties of the pasta. The present work evaluated the influence of the extract on the antioxidant capacity and microbial stability of the enriched pasta, as well as on its cooking properties.

## 2. Results and Discussion

### 2.1. Characterisation of Moringa oleifera Extract

In order to obtain a valuable ingredient for food fortification, a phenolic-rich extract was obtained from *Moringa oleifera* leaf powder (MOLP) using a solid–liquid extraction method with a Soxhlet extractor. Ethanol was chosen as solvent due to its polarity (that is suitable for phenolics’ extraction) and also because it is a Generally Recognised as Safe Substance (GRAS), which is fundamental for food applications. The phenolic-rich extract obtained was characterised regarding its total phenolic content (TPC) and anti-oxidant properties and the results are described in Table 3.

The MOLP extract presented a TPC of 79.0 ± 16.6 mg_GAE_/g. Regarding the anti-oxidant capacity, the extract was able to inhibit both radicals, being more effective for ABTS since a lower IC_50_ value was obtained when compared to DPPH (205.2 ± 4.6 mg/L vs. 636.0 ± 9.2 mg/L), meaning less extract is needed to reduce by half the free radical concentration. These parameters are greatly influenced by the plant origin and cultivation processes, as well as the extraction method and conditions [35,36]. A previous study used similar extraction conditions (Soxhlet extraction for 110 min with a sample-to-solvent ratio of 1:36 m/V) as the ones performed in the present work (Soxhlet extraction for 120 min with a sample-to-solvent ratio of 1:40 m/V), except for the extraction solvent (70% acetone vs. ethanol), and the extract presented a TPC of 26.07 ± 0.80 mg_GAE_/g [37]. These results may indicate that ethanol is more suitable to extract phenolic compounds from *M. oleifera* leaves. In another literature report, using an ultrasound-assisted solid–liquid extraction (solvent—ethanol; time—3 h; sample-to-solvent ratio—1:40 m/V), the MOLP extract presented a lower TPC (54.5 ± 16.8 mg_GAE_/g) but higher anti-oxidant capacity, with lower IC_50_ values (133.4 ± 12.3 mg/L for DPPH and 60.0 ± 9.9 mg/L for ABTS) [22]. Therefore, a higher TPC is not necessarily correlated with a higher anti-oxidant capacity.

Taking into consideration the interesting anti-oxidant properties presented by the extract, this was incorporated into fresh pasta. The effect of the addition of the extract on the anti-oxidant and cooking properties of the pasta, as well as on the product’s shelf-life, was evaluated.

### 2.2. Characterisation of Fortified Fresh Pasta

To analyse the possibility of using MOLP extract to create a value-added cereal-based product, different formulations of fresh pasta were prepared: a negative control without the incorporation of any additional ingredients (NC) and two formulations containing the MOLP extract at different wheat flour substitution levels—1.25% (P1.25) and 2% (P2). Figure 1 shows the different formulations produced. It is possible to observe that homogeneous pasta formulations were produced. Also, with the increasing content of MOLP extract, the fresh pasta presented a more green colour. 

To understand the impact of the incorporation of the MOLP extract on the pasta’s properties, the anti-oxidant capacity and cooking loss were analysed. From Figure 2, it is possible to observe that the addition of the *M. oleifera* extract increased the anti-oxidant properties of the fresh pasta and this was higher for ABTS radical, which is in accordance with the results obtained for the extract. Regarding DPPH assay (Figure 2A), no significant differences were observed between the antioxidant properties of P1.25 and P2 formulations and both remained relatively stable for two weeks. Regarding ABTS assay (Figure 2B), although P1.25 and P2 formulations presented similar inhibition percentages right after production (t_0_), the pasta containing less extract (P1.25) showed a higher decrease in the radical’s inhibition, throughout two weeks, when compared to the pasta containing a higher amount of extract (P2). No literature reports were found regarding the incorporation of *M. oleifera* extracts into fresh pasta. However, some authors have investigated the fortification of this food with MOLP and also observed an increase in the anti-oxidant capacity after production with the increasing level of fortification [32], although these properties were not analysed over time.

The cooking properties of pasta are an important aspect for the consumers’ acceptability of the product. For example, it is desirable that the solids that diffuse from the pasta structure to the cooking water are minimal. Therefore, the cooking loss was assessed to evaluate the cooking performance of pasta and the results are presented in Table 4. All formulations presented a cooking loss inferior to 8%, which is considered the acceptable technological limit for fresh pasta [32]. Also, no significant variations were observed in this parameter for two weeks. This is a crucial aspect, as maintaining its characteristics during storage is a desirable feature for food products. Finally, the addition of the MOLP extract slightly decreased the loss of solids during cooking. One study also reported cooking loss values inferior to 8% for fresh pasta fortified with MOLP. However, the incorporation of *M. oleifera* increased this parameter, although higher fortification percentages were tested (up to 15%) [32]. Other researchers have investigated the use of agro-industrial by-products to fortify pasta. They observed an increase in the antioxidant capacity upon food fortification however, the incorporation of these ingredients also increased the cooking loss [2]. Therefore, the use of extracts rather than the powder may present an advantage when it comes to maintaining the cooking properties of fresh pasta. 

Finally, the presence of yeast and moulds in the formulations was investigated two and eight days after production and the results are presented in Figure 3. As expected, taking into consideration the normal shelf-life of fresh pasta (2–3 days), no micro-organisms were detected two days after the production of the pasta (Figure 3A). Eight days after production (Figure 3B), it was observed the presence of micro-organisms in the formulation without *M. oleifera* extract (NC) but not in the fresh pasta containing the extract (P1.25 and P2). These results demonstrated that the incorporation of MOLP extract into fresh pasta prevents food spoilage by microbial organisms for longer periods of time, extending the shelf-life of the product.

Overall, the results obtained in the present work demonstrated the potential use of *M. oleifera* extracts as an ingredient to improve the antioxidant properties of fresh pasta, without affecting its cooking properties, and extend the food shelf-life. This study presents a biotechnological approach that uses natural compounds to create a value-added product with beneficial properties for human health and the food itself.

## 3. Materials and Methods

### 3.1. Samples and Reagents

The *Moringa oleifera* leaves were collected in Luanda, Angola (8°57′24.9′’ S, 13°13′02.9′’ E) and ground to obtain a homogeneous powder (particle size < 250 μm). To produce fresh pasta, wheat flour (type 55), medium size eggs, salt, and olive oil were purchased from a supermarket in Porto, Portugal.

To extract the phenolic compounds, ethanol (Ref. 83813.360, C_2_H_6_O, CAS 64-17-5) was acquired from VWR (Radnor, PA, USA). For the characterisation of the extract, sodium carbonate (Ref. 13418, CNa_2_O_3_, CAS 497-19-8) was obtained from Honeywell (Charlotte, NC, USA), while Folin–Ciocalteu reagent (Ref. 47641), 2,2′-azino-bis(3-ethylbenzothiazoline-6-sulfonic acid) (Ref. A1888, C_18_H_24_N_6_O_6_S_4_, CAS 30931-67-0) and 2,2-diphenyl-1-picrylhydrazyl (Ref. D9132, C1_8_H_12_N_5_O_6_, CAS 1898-66-4) were acquired from Sigma-Aldrich (St. Louis, MO, USA). For the microbial analysis, m-Lauryl Sulfate Broth (Ref. 0734) was obtained from Sigma-Aldrich, agar (Ref. J637, CAS 9002-18-0) was purchased from VWR and Rose-Bengal Chloramphenicol Agar (Ref. 1.00467.0500) was acquired from Merck (Darmstadt, Germany). Water purification was performed using a purification equipment (electrical resistance of 18.2 W) from Millipore (Burlington, MA, USA).

### 3.2. Extraction of Phenolic Compounds

The phenolic compounds present in the *Moringa oleifera* leaf powder (MOLP) were extracted using a solid–liquid extraction with a Soxhlet apparatus. The extraction was performed for 2 h, using ethanol as solvent, in a sample-to-solvent ratio of 1:40 m/V. Solvent evaporation was performed in a rotary evaporator (Rotavapor R-200, BUCHI Laboratories, Flawil, Switzerland), followed by a constant stream of nitrogen (2 mbar).

### 3.3. Characterisation of Moringa oleifera Extract

The total phenolic content (TPC) of the extract was determined using the Folin–Ciocalteu method, according to the literature [22]. The extract solution (1 g/L) was incubated with the Folin–Ciocalteu reagent and sodium carbonate solution (333.3 g/L) for 2 h at room temperature. The absorbance was analysed at 750 nm and the results were expressed in mg of gallic acid equivalents (GAE)/g of extract.

The 2,2-diphenyl-1-picrylhydrazyl (DPPH) and 2,2′-azino-bis(3-ethylbenzothiazoline-6-sulfonic acid) (ABTS) assays were performed to determine the anti-oxidant capacity of the MOLP extract, according to the literature with slight modifications [22]. For the ABTS assay, the extract solutions (1.5–8.0 g/L) were incubated with the ABTS solution for 15 min and the absorbance was analysed at 734 nm. For the DPPH assay, the extract solutions (0.1–2.5 g/L) were incubated with the DPPH solution for 40 min and the absorbance was analysed at 515 nm. In both assays, the percentage of inhibition of the free radicals was calculated and the IC_50_ values were determined.

### 3.4. Incorporation of Moringa oleifera Extract in Fresh Pasta

#### 3.4.1. Production of Fresh Pasta

To evaluate the effect of the incorporation of MOLP in the fresh pasta, three formulations were prepared: negative control (NC)—no additional ingredients; fresh pasta with MOLP at a wheat flour substitution level of 1.25% *w*/*w* (P1.25); and fresh pasta with MOLP at a wheat flour substitution level of 2% *w*/*w* (P2). The ingredients of each formulation are detailed in Table 5. Briefly, wheat flour (type 55), eggs (medium size; average weight: 50 g), salt, and olive oil were kneaded by hand until a homogeneous mixture was obtained. The dough was reserved for 30 min at 4 °C. Afterwards, the dough was flattened and cut using a home-scale pasta machine (Kmt^®^Style, China) to obtain pasta with 6 mm width and 2 mm thickness. All formulations were kept at 4 °C inside a plastic container, protected from air and light, until further analysis.

#### 3.4.2. Anti-oxidant Capacity

The anti-oxidant properties of the fresh pasta were analysed for two weeks. For that, the phenolic compounds present in each formulation were extracted using ethanol. Briefly, 4 mL of ethanol were added to 2 g of fresh pasta and the extraction procedure (1 min of vortex followed by 5 min in an ultrasonic bath) was performed three times. The solutions were centrifuged for 20 min at 3000 rpm using a Rotofix 32 A centrifuge (Hettich, Tuttlingen, Germany). The supernatant was collected and, again, 4 mL of ethanol were added to the fresh pasta and the whole procedure was repeated once more. The supernatant was used to analyse the anti-oxidant capacity (ABTS and DPPH) of the different formulations according to the methods described in Section 3.3.

#### 3.4.3. Cooking Loss

The cooking loss (CL) of each formulation was evaluated for two weeks according to the literature, with some modifications [38]. About 5 g of fresh pasta were cooked in 100 mL of boiling water for 5 min. The cooked pasta was removed from the cooking water, rinsed, and drained for 5 min before being weighed. The cooking and rising water were collected into a beaker and evaporated in a convection oven at 105 °C to obtain the cooking residue. Finally, the CL was determined using Equation (1)
CL (%) = (m_residue_/m_raw_) × 100,(1)
where m_residue_ is the mass of the cooking residue and m_raw_ is the mass of the fresh pasta before cooking.

#### 3.4.4. Microbial Analysis

The presence of yeast and moulds in the fresh pasta was analysed using Rose Bengal Chloramphenicol Agar (RBC). The fresh pasta was placed in a saline solution (0.9% NaCl) and vortexed. Afterwards, 100 μL of each sample was plated and the RBC plates were incubated at 25 °C for 7 days. Finally, the presence of micro-organisms on the plates was analysed. 

### 3.5. Statistical Analysis

The statistical analysis was performed in GraphPad Prism 8.0.2 with an analysis of variance (ANOVA), using Tukey’s multiple comparisons test. Values with *p* < 0.05 (95% confidence interval) were considered statistically different.

## 4. Conclusions

This work intended to assess the possibility of using extracts rich in phenolic compounds, known for their anti-oxidant capacity, in a sustainable fashion, to increase the value of cereal-based products, more particularly fresh pasta. The extract obtained from *Moringa oleifera* leaf powder presented anti-oxidant properties, which may extend the product’s shelf-life while contributing to human health. The fresh pasta incorporated with *M. oleifera* extract presented higher anti-oxidant capacity when compared to the negative control. Also, the cooking properties, specifically the cooking loss, were not impaired by the addition of the extract. Finally, eight days after production, microbial contamination was only detected in the negative control formulation and not in the fortified pasta, meaning that the phenolic-rich extract was able to extend the shelf-life of the product. These results demonstrate the potentiality of food fortification and present a possible application of a novel approach to the food industry to produce value-added products, especially carbohydrate-rich foods that lack nutritional value. The present study is a proof of concept for the possibility to use *M. oleifera* extracts to fortify fresh pasta; however, more studies are recommended to fully comprehend the effect of the extract’s addition on the final product. First, toxicity studies of the obtained extract are crucial to ensure its safety for food applications. A nutritional evaluation of pasta must be performed, both before and after the incorporation of the extract, to understand the effect on its nutritional value. Also, higher substitution levels can be evaluated to determine the technical limit for the supplementation level and the effects on the nutritional and biological properties. The final shelf-life of the product should also be determined. Finally, it is important to conduct a sensory analysis to understand how the addition of *M. oleifera* extract impacts the sensory properties of fresh pasta.

## Figures and Tables

**Figure 1 ijms-24-12451-f001:**
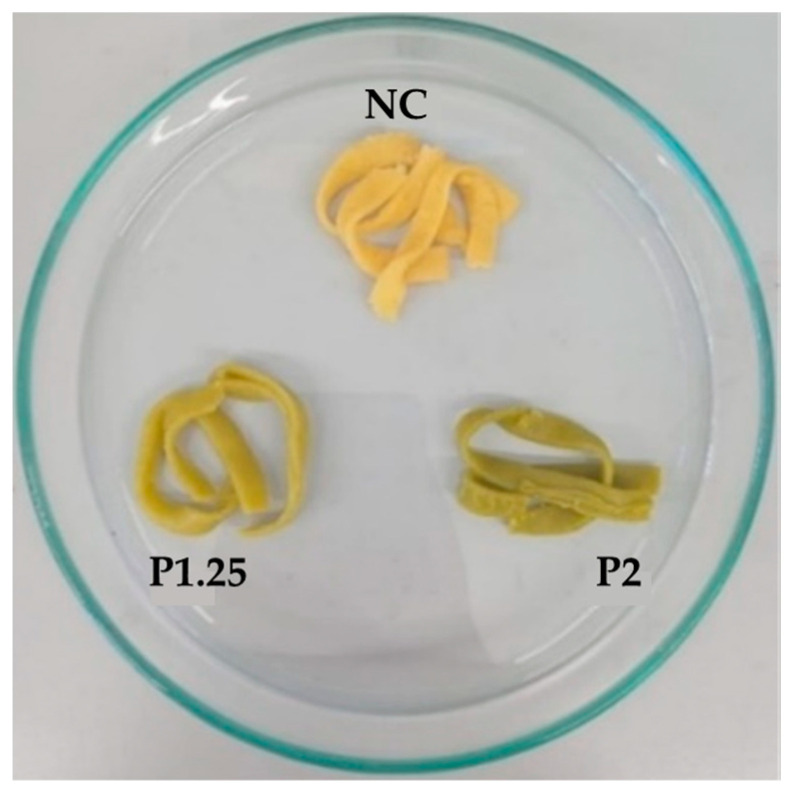
Fresh pasta formulations after production. NC—fresh pasta with no additional ingredients (negative control); P1.25—fresh pasta with 1.25% MOLP; P2—fresh pasta with 2% MOLP.

**Figure 2 ijms-24-12451-f002:**
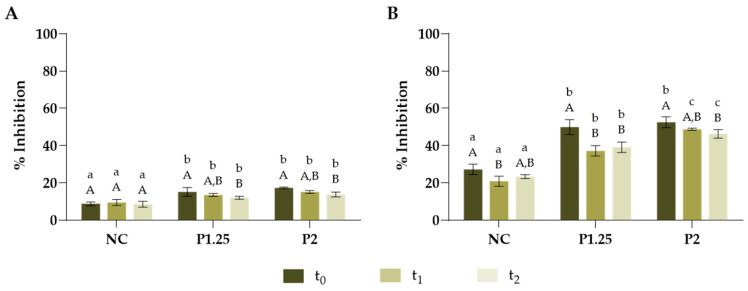
Fresh pasta’s antioxidant capacity over time for DPPH (**A**) and ABTS (**B**) assays. The analysis was performed at different timepoints: t_0_—after production; t_1_—one week after production; and t_2_—two weeks after production. NC—fresh pasta with no additional ingredients (negative control); P1.25—fresh pasta with 1.25% MOLP; P2—fresh pasta with 2% MOLP. The results are expressed as means ± standard deviations of 3 independent measurements. Different lowercase letters (a–c) represent statistically different values (*p* < 0.05) for different formulations at the same timepoint. Different capital letters (A–B) represent statistically different values (*p* < 0.05) for the same formulation at different timepoints.

**Figure 3 ijms-24-12451-f003:**
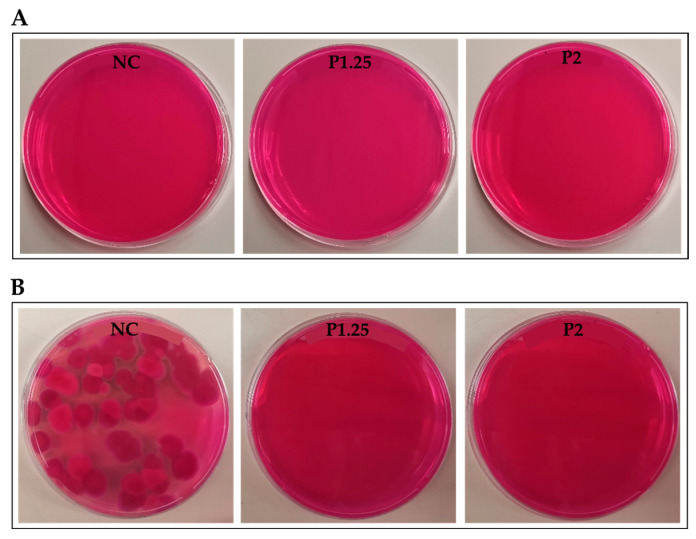
Presence of microorganisms in fresh pasta two days after production (**A**) and eight days after production (**B**). NC—fresh pasta with no additional ingredients (negative control); P1.25—fresh pasta with 1.25% MOLP; P2—fresh pasta with 2% MOLP.

**Table 1 ijms-24-12451-t001:** Examples of data found in the literature on the toxicity of *Moringa oleifera* extracts.

Sample	System	Treatment Plan	Results	Ref.
Aqueous extract from MOLP	Rats	Single dose (400–2000 mg/kg)	No mortality was recorded at any concentration. The results showed that under 2000 mg/kg, the plant is safe for consumption and medical use.	[14]
		Daily administration (400–1600 mg/kg for 21 days)	Although at different doses the extract caused significant changes in the levels of total proteins and liver enzymes, organ pathology did not reveal any significant lesion. The study concluded that the extract demonstrated a relatively safe profile.	
	Human PBMC	Single dose (5–80 mg/mL)	Concentrations above 20 mg/mL proved to be cytotoxic, as it was recorded an increase in the total amount of LDH released. However, this is a concentration not achievable by oral ingestion.	[15]
Methanol extract from *M. oleifera* seeds	Rats	Single dose (1000–5000 mg/kg)	Although signs of acute toxicity were observed at doses of 4000 mg/kg, no side effects were detected at concentrations lower than 3000 mg/kg.	[16]
		Daily administration (400–1600 mg/kg for 21 days)	At 1600 mg/kg, the extract led to an increase in the ALT and AST levels and a decrease in the weight of the rat. However, the study concluded that the extract is safe for both medical and nutritional applications.	
Ethanol extract from MOLP	Mice	Single dose (200–6400 mg/kg)	Although it was detected signs of sedation and reduced locomotion at high doses for 2 h after administration, the extract did not produce any mortality up to 6400 mg/kg.	[17]

MOLP—*Moringa oleifera* leaf powder; PBMC—peripheral blood mononuclear cells; LDH—lactate dehydrogenase; ALT—alanine transferase; AST—aspartate transferase.

**Table 2 ijms-24-12451-t002:** Studies on the incorporation of *Moringa oleifera* in cereal-based food products.

Product	Objective	Results	Ref.
Cake	Study the nutritional composition of *M. oleifera*-supplemented wheat cake and evaluate its acceptability.	The addition of MOLP (up to 20%) increased the protein, fibre, and ash content of the cake while reducing its carbohydrate and fat content. However, higher levels of *M. oleifera* addition decreased its sensory acceptance. The best results were obtained at 8% of MOLP addition.	[26]
Biscuits	Evaluate the nutritional benefits and acceptance among consumers of the incorporation of MOLP in biscuits.	Supplementation with MOLP enhanced the nutritional value of the biscuits (increased iron and protein content). However, it affected the physical properties and acceptance of the biscuits, mainly in terms of colour and texture.	[27]
	Study the quality characteristics and organoleptic attributes of biscuits substituted with MOLP.	The incorporation of *M. oleifera* significantly increased the protein, iron, and calcium content of the biscuits. Biscuits incorporated with 10% *M. oleifera* presented acceptable sensorial properties. Above this level, the supplementation negatively affected their sensorial acceptability.	[28]
Bread	Evaluate the effect of the supplementation of wheat bread with MOLP on its physicochemical and sensory properties.	*M. oleifera* addition increased the bread’s fibre, ash, and protein content and weight while decreasing its moisture and loaf volume. Higher levels of MOLP substitution drastically reduced the acceptability of the bread.	[29]
	Determine the effect of the fortification of bread with MOLP on its quality and nutritional value.	Bread samples became darker as the concentration of *M. oleifera* increased, whilst nutrient levels increased. The overall consumer acceptability of the bread decreased with increasing supplementation. Bread supplemented with 5% MOLP achieved the best results.	[30]
	Assess the physical, sensorial and antioxidant properties of gluten-free bread enriched with MOLP.	The fortified bread presented an increase in the phenolic content and antioxidant activity. The bread with higher acceptability was obtained at 2.5% of *M. oleifera* substitution.	[31]
Pasta	Fortify wheat fresh pasta with MOLP to increase its nutritional value.	The enriched pasta presented a higher phenolic content and anti-oxidant activity and good sensory acceptability. The fortification led to a decrease in the optimum cooking time, swelling index, and firmness while increasing its cooking loss and adhesiveness.	[32]
	Evaluate the potential use of MOLP in the fortification of pasta to be used as a natural immune booster.	Supplementation of pasta with MOLP improved its nutritional quality, increasing its content of bioactive compounds. The results indicated that pasta fortified with different concentrations of *M. oleifera* is well accepted and the cooking quality was not significantly affected.	[33]

MOLP—*Moringa oleifera* leaf powder.

**Table 3 ijms-24-12451-t003:** Results of the characterisation of *Moringa oleifera* extract.

TPC (mg_GAE_/g_extract_)	Antioxidant Capacity IC_50_ (mg/L)	
	DPPH	ABTS
79.0 ± 16.6	636.0 ± 9.2	205.2 ± 4.6

TPC—total phenolic content; GAE—gallic acid equivalents; IC_50_—concentration of extract necessary to inhibit 50% of the free-radical; DPPH—2,2-diphenyl-1-picrylhydrazyl; ABTS—2,2′-azino-bis(3-ethylbenzothiazoline-6-sulfonic acid). The results are expressed as means ± standard deviations of three independent measurements.

**Table 4 ijms-24-12451-t004:** Fresh pasta’s cooking loss variation over time.

Formulation	Cooking Loss (%)		
	t_0_	t_1_	t_2_
NC	2.14 ± 0.28 ^a^	2.39 ± 0.10 ^a^	2.06 ± 0.09 ^a^
P1.25	1.70 ± 0.06 ^b^	1.58 ± 0.04 ^b^	1.47 ± 0.23 ^b^
P2	1.99 ± 0.16 ^a,b^	1.77 ± 0.13 ^b^	1.86 ± 0.20 ^a,b^

NC—fresh pasta with no additional ingredients (negative control); P1.25—fresh pasta with 1.25% MOLP; P2—fresh pasta with 2% MOLP. The analysis was performed after production (t_0_), one week after production (t_1_), and two weeks after production (t_2_). The results are expressed as means ± standard deviations of 3 independent measurements. Different lowercase letters (a–b) represent statistically different values (*p* < 0.05) in the same column.

**Table 5 ijms-24-12451-t005:** List of ingredients of each fresh pasta formulation.

Ingredient	NC	P1.25	P2
Wheat flour (g)	100.00	98.75	98.00
No. Eggs	1	1	1
Salt (g)	0.20	0.20	0.20
Olive oil (g)	0.70	0.70	0.70
MOLP (g)	-	1.25	2.00

MOLP—*Moringa* oleifera leaf powder; NC—fresh pasta with no additional ingredients (negative control); P1.25—fresh pasta with 1.25% MOLP; P2—fresh pasta with 2% MOLP.

## Data Availability

The data presented in this study are available on request from the corresponding author.

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
