# Peer review of "Innovative Approaches for Food: Using Natural Phenolic-Rich Extracts to Produce Value-Added Fresh Pasta"

_ijms, 2023, doi:10.3390/ijms241512451_

Round 1

Reviewer 1 Report

The article titled " Biotechnological Approaches for food: using natural phenolic-
rich extracts to produce value-added fresh pasta" presents the characterization and value addition to wheat products through Moringa extract addition. The authors also evaluated the antioxidative potential of the product and assessed its shelf-life. The study looks containing some scientific merit while some major issues (as follows) are important to be addressed:

1.     What nutrients do lack them? Do cereal products contain only carbohydrates?

2.     The antioxidant test suffers from the use of a reference antioxidant agent to compare the potential of Moringa in the context of scavenging effects.

3.     Why didn't the author conduct the same assay for the finished product?

4.     The nutritional assessment of pre- and post-added nutritional values of pasta

5.     No proximate analysis is done in this study which experiences a great shortcoming of the study product and its nutritional values (carbon, moisture, ash, carb, protein, fat) because the authors started the abstract with the benefits of the proposed value-added food. So, the food value before and after needs to be addressed.

6.     Total antioxidant capacity and radical scavenging effects of finished products in the context of IC50 values are highly relevant as the authors accomplished for Moringa extract.

7.     In the results and discussion the authors justified the use of ethanol with the reference “A previous study used similar extraction conditions (Soxhlet extraction for 110 min with a sample-to-solvent ratio of 1:36 m/V) as the ones performed in the present work (Soxhlet extraction for 120 min with a sample-to-solvent ratio of 1:40 m/V), except for the extraction solvent (70% acetone vs. ethanol), and the extract presented a TPC of 26.07 ± 0.80 mgGAE/g”. This sort of extraction is not acceptable for food supplementation because there is acetone as well. Is it the right justification for using ethanol?

8.     Why did you use organic extract for food products? The extract toxicity is not checked in animal models. Why the aqueous/hydroalcoholic extract was not used? The extract itself needs to be checked for its toxicity in different models. The use of ethanol is even questionable until and unless the toxicity test of the extract ensures the nontoxic behavior of the extract. At least hydroalcoholic extract would be preferred. Although the authors referred to the toxicity in the table, but the toxicity may be varied based on the components added to the plant extract.  Authors should explain the issue because the solvent residue is also a matter.

9.      How did the authors select the 1.25 and 2.0% for supplementation? What is the basis of the percentage selection? Importantly, the doses have no geometrical relationship.

10.  What is the final shelf-life of the product?

11.  The composition shows the incorporation of medium size eggs which may vary from place to place/country to country. It's better to take the average weight of the eggs and mention their size in grams.

12.  How do the authors measure the absorbance of DPPH free radical scavenging effects using the wavelength 515nm ( do you have any reference)? So far, its established as 517nm.

13.  Please address the limitations of the study in the conclusion part and explain how can you overcome those?

Minor comments

1. Is it a biotechnological approach? Rethink and revise the title. In a true sense, no biotechnological approach exists.

2. In Figure 2, the color of t1 and t2 should be more distinctive

3. The organoleptic change of the prepared pasta is not monitored. It’s highly important form food products. However, different pasta looks different color.

File attached

Author Response

Comments and Suggestions for Authors

The article titled " Biotechnological Approaches for food: using natural phenolic-rich extracts to produce value-added fresh pasta" presents the characterization and value addition to wheat products through Moringa extract addition. The authors also evaluated the antioxidative potential of the product and assessed its shelf-life. The study looks containing some scientific merit while some major issues (as follows) are important to be addressed:

  1. What nutrients do lack them? Do cereal products contain only carbohydrates?

Answer: Although cereal products contain other nutrients, like proteins for example, the major nutrient present in this type of product is carbohydrates. The authors give some examples of the nutrients that they lack (lines 32-33).

  1. The antioxidant test suffers from the use of a reference antioxidant agent to compare the potential of Moringa in the context of scavenging effects.

Answer: Thank you for your comment. The authors will take your advice into consideration in future work.

  1. Why didn't the author conduct the same assay for the finished product?

Answer:  The authors opted to perform the same assays for the extract and the finished product. Therefore, the assay using a reference antioxidant agent was not performed for fresh pasta.

  1. The nutritional assessment of pre- and post-added nutritional values of pasta.

Answer: Although the authors understand the importance of performing a nutritional assessment of pasta before and after fortification, the main focus of this work was to evaluate the antioxidant properties and the possibility to extend the shelf-life of the product. Nevertheless, as the authors believe that the nutritional evaluation is fundamental to understand if the incorporation of Moringa oleifera extracts does improve the nutritional value of pasta, they plan to do it in future research. A brief sentence was added to the conclusion section about future work (lines 298-300).

  1. No proximate analysis is done in this study which experiences a great shortcoming of the study product and its nutritional values (carbon, moisture, ash, carb, protein, fat) because the authors started the abstract with the benefits of the proposed value-added food. So, the food value before and after needs to be addressed.

Answer: Thank you for your comment. As mentioned in the abstract, the addition of Moringa oleifera extracts may present a strategy to improve the nutritional and biological value of the product. In the present work, the authors opted to focus on the biological properties of pasta, more specifically their antioxidant properties, instead of the nutritional value since, in the literature, the nutritional composition is more studied than the antioxidant potential. However, the authors understand that it is extremely important to study the effect of the extract on the nutritional profile of the product and they plan to do it in future research. A brief sentence was added to the conclusion section about future work (lines 298-300).

  1. Total antioxidant capacity and radical scavenging effects of finished products in the context of IC50 values are highly relevant as the authors accomplished for Moringa extract.

Answer: Thank you for your comment. While different concentrations of the extract were analysed, in the case of the finished product only the product itself was evaluated. Therefore, for the final product, only a percentage of inhibition was obtained, which does not allow the determination of the IC50 value. The authors believe that, since pasta is not consumed in solution, it was more relevant to analyse only the product itself.

  1. In the results and discussion the authors justified the use of ethanol with the reference “A previous study used similar extraction conditions (Soxhlet extraction for 110 min with a sample-to-solvent ratio of 1:36 m/V) as the ones performed in the present work (Soxhlet extraction for 120 min with a sample-to-solvent ratio of 1:40 m/V), except for the extraction solvent (70% acetone vs. ethanol), and the extract presented a TPC of 26.07 ± 0.80 mgGAE/g”. This sort of extraction is not acceptable for food supplementation because there is acetone as well. Is it the right justification for using ethanol?

Answer: Thank you for your comment. The authors intended to demonstrate that the TPC of an extract can be influenced by numerous parameters, in this case, the extraction solvent, without taking into consideration its final application. Of course, when extracts are obtained for a defined application (like in this work), parameters such as the solvent used to extract the compounds of interest need to be carefully thought. Using this reference, the authors did not intend to justify why ethanol was chosen in this work but only to demonstrate that the TPC can be influenced by the extraction solvent (although acetone could never be used for food supplementation applications).

  1. Why did you use organic extract for food products? The extract toxicity is not checked in animal models. Why the aqueous/hydroalcoholic extract was not used? The extract itself needs to be checked for its toxicity in different models. The use of ethanol is even questionable until and unless the toxicity test of the extract ensures the nontoxic behavior of the extract. At least hydroalcoholic extract would be preferred. Although the authors referred to the toxicity in the table, but the toxicity may be varied based on the components added to the plant extract. Authors should explain the issue because the solvent residue is also a matter.

Answer: Thank you for your comment. The authors opted for an alcoholic extract, instead of an aqueous/hydroalcoholic extract, because ethanol is simpler to evaporate than water. While for the evaporation of ethanol, the utilisation of a rotary evaporator is sufficient, to evaporate the water it would be necessary to use lyophilisation. This process requires more time, energy and money, which the authors believe could pose a disadvantage to the process. Although it was outside of the scope of this work, the authors intend to perform cytotoxicity analysis on the extracts, to ensure the consumer’s safety. A brief sentence was added to the conclusion section about future work (lines 294-304).

  1. How did the authors select the 1.25 and 2.0% for supplementation? What is the basis of the percentage selection? Importantly, the doses have no geometrical relationship.

Answer: The authors intended to evaluate the effect of different substitution levels on the properties of fresh pasta. However, due to the availability of the raw material (Moringa oleifera powder) at the time, little amount of extract was obtained and it was not possible to study higher supplementation levels, as intended. Nevertheless, the results with these substitution levels were already quite promising and the authors intend to study higher levels in future work, to understand what is the technological limit for the supplementation level, as well as the effect on the sensory properties. A brief sentence was added to the conclusion section about future work (lines 294-304).

  1. What is the final shelf-life of the product?

Answer: The authors evaluated the shelf-life of the product through the evaluation of the presence of microorganisms in the product. That analysis was performed approximately one week after production. Two weeks after production, when analysing the antioxidant properties, the authors did not detect any microorganisms visually. However, due to time constraints, it was not possible to corroborate their absence with microbiological methods. The authors understand that this is a limitation of the present study and there is something that needs to be considered for future work. A brief sentence about this topic was added to the conclusion section (line 302).

  1. The composition shows the incorporation of medium size eggs which may vary from place to place/country to country. It's better to take the average weight of the eggs and mention their size in grams.

Answer: Thank you for your comment. The information was added to the manuscript, as suggested (lines 243-244).

  1. How do the authors measure the absorbance of DPPH free radical scavenging effects using the wavelength 515nm ( do you have any reference)? So far, its established as 517nm.

Answer: The choice of the wavelength was based on the literature: Bobo-García, G.; Davidov-Pardo, G.; Arroqui, C.; Vírseda, P.; Marín-Arroyo, M.R.; Navarro, M. Intra-laboratory validation of microplate methods for total phenolic content and antioxidant activity on polyphenolic extracts, and comparison with conventional spectrophotometric methods. J. Sci. Food Agric. 2014, 95, 204–209. https://doi.org/10.1002/jsfa.6706.

  1. Please address the limitations of the study in the conclusion part and explain how can you overcome those?

Answer: A brief paragraph was added to the conclusion section about the limitations and future work, as suggested (lines 294-304).

Minor comments

  1. Is it a biotechnological approach? Rethink and revise the title. In a true sense, no biotechnological approach exists.

Answer: Thank you for your comment. Upon revision, the authors altered the title to “Innovative approaches for food: using natural phenolic-rich extracts to produce value-added fresh pasta”.

  1. In Figure 2, the color of t1 and t2 should be more distinctive.

Answer: the authors altered the colours of t1 and t2 in Figure 2, as suggested.

  1. The organoleptic change of the prepared pasta is not monitored. It’s highly important form food products. However, different pasta looks different color.

Answer: Thank you for your comment. Five members of the laboratory have tasted the pasta and the differences in the taste were minimal. However, this test was performed more as a curiosity and no proper sensory evaluation analyses were performed. Therefore, no data regarding the sensory characteristics of pasta were displayed in the manuscript. Nevertheless, the authors understand the importance of performing a sensory evaluation and plan to do it in future research. A brief sentence was added to the conclusion section about future work (lines 302-304).

Reviewer 2 Report

Dear Authors,

the paper is interesting and current.

I think it is necessary to make some changes and insert missing information to improve what is written.

My comments below:

Introduction

Tables you have inserted are very interesting but it is necessary to provide further information on the difference between the extract you have used in this study and that used by other authors.

It is not clear what the differences are, what is better, etc.

L. 39: Moringaceae in italic?

L. 68: It is better to talk about a food ingredient and not a food additive for regulatory reasons.

Figure 1. Delete “no additive”.

Results and discussion

L. 144-146: “However, some authors have investigated the fortification of this food with MOLP and also observed an increase in the antioxidant capacity after production with the increasing level of fortification [32], although these properties were not analysed over time”. Apparently this sentence contradicts itself. Specify what type of MOLP the other authors used

L. 152: Delete “no additive”.

L. 175: Delete “no additive”.

L. 191: Delete “no additive”.

L. 180-192 and Figure 3: The picture of the Petri dishes is very easy to understand but you should also enter the average value of the cfu/g or, alternatively, the % of positive samples.

Just for my curiosity: is the taste of the pasta different? Sensory evaluation is an important aspect when intending to enrich a food.

Materials and methods

L. 202: “medium eggs”: it is advisable to also indicate the size for those who do not know the European legislation.

L. 238: Delete “no additive”.

L. 215: temperature?

L. 245: “All formulations were kept at 4 °C until further analysis.” Describe the method of packaging (plastic bag? Exposed to the air…) and storage. 

Conclusions

L. 285: delete with no additive.

Author Response

Comments and Suggestions for Authors

Dear Authors, the paper is interesting and current. I think it is necessary to make some changes and insert missing information to improve what is written. My comments below.

Introduction

Tables you have inserted are very interesting but it is necessary to provide further information on the difference between the extract you have used in this study and that used by other authors. It is not clear what the differences are, what is better, etc.

Answer: Thank you for your comment. In the present work, the extract obtained from Moringa oleifera leaves is incorporated into fresh pasta. In other studies, as exemplified in Table 2 and mentioned in lines 78-80 of the manuscript, no extract is used, only the Moringa oleifera leaf powder (MOLP). Therefore, this is the first article using the extract to fortify pasta. However, the authors believe that the use of the extract may present an advantage since the bioactive compounds are more concentrated than in the powder, which may allow the incorporation of higher amounts of the compounds of interest.

  1. 39: Moringaceae in italic?

Answer: The authors believe that, for plants, scientific names of taxa above the genus level (like family) are not written in italic.

  1. 68: It is better to talk about a food ingredient and not a food additive for regulatory reasons.

Answer: Thank you for your comment. The suggested alteration was made (line 69).

Figure 1. Delete “no additive”.

Answer: The authors substituted “no additive” for “no additional ingredients” (line 132).

Results and discussion

  1. 144-146: “However, some authors have investigated the fortification of this food with MOLP and also observed an increase in the antioxidant capacity after production with the increasing level of fortification [32], although these properties were not analysed over time”. Apparently this sentence contradicts itself. Specify what type of MOLP the other authors used.

Answer: Thank you for your comment. What the authors intended to explain was that although no studies were found in the literature about the incorporation of the extract obtained from Moringa oleifera leaves, some evaluated the incorporation of the leaves powder (Moringa oleifera leaves powder – MOLP). In the referred study, they observed an increase in the antioxidant properties of the fresh pasta containing the powder upon production (compared to the pasta without the powder) but this was not analysed over time, as in this study.

  1. 152: Delete “no additive”.

Answer: The authors substituted “no additive” for “no additional ingredients” (line 154).

  1. 175: Delete “no additive”.

Answer: The authors substituted “no additive” for “no additional ingredients” (line 177).

  1. 191: Delete “no additive”.

Answer: The authors substituted “no additive” for “no additional ingredients” (line 193).

  1. 180-192 and Figure 3: The picture of the Petri dishes is very easy to understand but you should also enter the average value of the cfu/g or, alternatively, the % of positive samples.

Answer: Thank you for your comment. As seen in Figure 3, the colonies present in the Petri dish were not completely separated from each other, making it difficult to count them precisely. Therefore, the authors opted to not determine the CFU/g for believing that that measurement could present significant errors. Moreover, all samples analysed (for NC formulation) presented contaminations.

Just for my curiosity: is the taste of the pasta different? Sensory evaluation is an important aspect when intending to enrich a food.

Answer: Thank you for your comment. Five members of the laboratory have tasted the pasta and the differences in the taste were minimal. However, this test was performed more as a curiosity and, therefore, no proper sensory evaluation analyses were performed. This article intended to be a proof of concept for the use of M. oleifera extracts as ingredients to extend the fresh pasta’s shelf-life and the sensory analysis was not inside the scope of the present work. Nevertheless, the authors understand the importance of performing a sensory evaluation and plan to do it in future research. A brief sentence was added to the conclusion section about future work (lines 302-304).

Materials and methods

  1. 202: “medium eggs”: it is advisable to also indicate the size for those who do not know the European legislation.

Answer: Thank you for your comment. The authors added the average weight of “medium eggs” to the manuscript (lines 243-244).

  1. 238: Delete “no additive”.

Answer: The authors substituted “no additive” for “no additional ingredients” (line 240).

  1. 215: temperature?

Answer: The extraction of the phenolic compounds was performed at the temperature necessary for the ethanol to boil (boiling point of ethanol: 78 °C.

  1. 245: “All formulations were kept at 4 °C until further analysis.” Describe the method of packaging (plastic bag? Exposed to the air…) and storage.

Answer: The authors added some information about the packaging and storage to the manuscript (lines 247-248).

Conclusions

  1. 285: delete with no additive.

Answer: The alteration was made, as suggested (line 288).

Round 2

Reviewer 1 Report

Authors have addressed the queries

Commented above

Reviewer 2 Report

Dear Authors,

the revised paper responded to the requests.

Best regards